# Moist bias in the Pacific upper troposphere and lower stratosphere (UTLS) in climate models affects regional circulation patterns

Felix Ploeger[1,2], Thomas Birner[3], Edward Charlesworth[1], Paul Konopka[1], and Rolf Müller[1]

[1]Institute for Energy and Climate Research: Stratosphere (IEK–7), Forschungszentrum Jülich, Jülich, Germany.
[2]Institute for Atmospheric and Environmental Research, University of Wuppertal, Wuppertal, Germany.
[3]Meteorological Institute Munich, Ludwig Maximilians University of Munich, Munich, Germany.

**Correspondence:** Felix Ploeger (f.ploeger@fz-juelich.de)

**Abstract.** Water vapour in the UTLS is a key radiative agent and a crucial factor in the Earth's climate system. Here, we investigate a common regional moist bias in the Pacific UTLS during northern summer in state-of-the-art climate models. We demonstrate, through a combination of climate model experiments and satellite observations that the Pacific moist bias amplifies local longwave cooling which ultimately impacts regional circulation systems in the UTLS. Related impacts involve a strengthening of isentropic potential vorticity gradients, strengthened westerlies in the Pacific westerly duct region, and a zonally displaced anticyclonic monsoon circulation. Furthermore, we show that the regional Pacific moist bias can be significantly reduced by applying a Lagrangian, less diffusive transport scheme and that such a model improvement could be important for improving the simulation of regional circulation systems, in particular in the Asian monsoon and Pacific region.

## 1 Introduction

Notwithstanding the low abundance of water vapour in the stratosphere, with mixing ratios on the order of parts per million above the cold tropical tropopause (Brewer, 1949; Fueglistaler et al., 2005), stratospheric water vapour variations play a crucial role for climate variability. Decadal variations in stratospheric water vapour have been shown to modify the radiative budget on a decadal scale by up to 30% (Solomon et al., 2010). In a future climate with rising greenhouse gas levels, climate models predict significant stratospheric water vapour increases, strongest in the lowermost stratosphere directly above the extratropical tropopause (Dessler et al., 2013; Banerjee et al., 2019). The processes causing this increase are under debate (Dessler et al., 2016; Smith et al., 2022; Hoor et al., 2010). Recently, indications for an increase in stratospheric water vapour have been found from satellite observations (Konopka et al., 2022) which could be partly related to decadal variability (Tao et al., 2023). As stratospheric water vapour acts as an impactful greenhouse gas, an increase in its abundance will cause a positive climate feedback, ranging between about 0.1-0.3 $Wm^{-2}K^{-1}$ (Banerjee et al., 2019; Dessler et al., 2013; Nowack et al., 2023), and substantial effects on atmospheric temperatures and dynamics (Li and Newman, 2020), with the largest fraction of these effects caused by water vapour increases in the lowermost stratosphere.

However, it is particularly the lowermost stratosphere where current global climate models show the largest differences between simulations and observations in stratospheric water vapour (Keeble et al., 2021). These moist biases in the simulations appear to be related to numerical diffusion in model transport schemes; they can be significantly reduced when employing

less diffusive Lagrangian schemes (Stenke et al., 2009; Charlesworth et al., 2023). Charlesworth et al. (2023) showed that lower stratospheric water vapour, in particular the model moist bias in that region, exerts a first order effect on the zonal mean atmospheric circulation, with water vapour increases causing a strengthening of the stratospheric circulation, upward and poleward shifts of the subtropical jets, and a poleward shift of the tropospheric eddy-driven jet. Hence, improving the representation of stratospheric water vapour in climate models appears to be essential for improving the simulation of zonal mean circulation in climate models and future projections.

A particularly strong moisture source for the lower stratosphere is the anticyclonic Asian monsoon circulation during boreal summer (Randel and Park, 2006; Garny and Randel, 2013), which causes the ascending moist air masses to partly bypass the coldest regions of the tropical tropopause and to retain relatively high moisture (James et al., 2008; Wright et al., 2011; Nützel et al., 2019). Hence, the Asian monsoon circulation features the largest positive water vapour anomaly in the UTLS (here, "anomaly" is used for the deviation from zonal mean while "bias" is used for the difference between model and observations). Although the air masses are, to some degree, confined within the monsoon anticyclone (Park et al., 2009; von Hobe et al., 2021; Legras and Bucci, 2020), Rossby-wave breaking along the anticyclone edge may cause substantial transport of moisture into the extratropical lowermost stratosphere (Ploeger et al., 2013; Rolf et al., 2018). Analysis of the distribution of water vapour in the lower stratosphere simulated by a specific climate model has shown a substantial model bias in the Asian monsoon region (Wang et al., 2018). In the model, the maximum monsoon moisture anomaly was found to be displaced far into the Pacific region, compared to satellite observations which show the moisture maximum in the Asian monsoon circulation. Wang et al. (2018) further showed that this model bias was likely a transport issue and could be reduced by increasing the model resolution.

Here, we combine a dedicated climate model experiment with the ECHAM MESSy Atmospheric Chemistry model (EMAC), statistical analysis of a suite of historical climate model simulations from the Coupled Model Intercomparison Project (CMIP6), and satellite observations by the Microwave Limb Sounder (MLS) to investigate regional water vapour anomalies in the UTLS and related effects on regional circulation systems. The EMAC model experiment employs either the standard "control" config-uration or a new "modified–Lagrangian" transport scheme with the Chemical Lagrangian Model of the Stratosphere (CLaMS) coupled to EMAC (Methods), allowing the effects of model transport on stratospheric water vapour to be isolated. These EMAC model simulations are the same as those recently published by Charlesworth et al. (2023), but here we extend their zonal mean analysis to a regional level to address the following questions: (i) Where are the strongest biases in UTLS water vapour in current climate models compared to satellite observations? (ii) How do regional anomalies in UTLS water vapour affect regional atmospheric circulation systems, in particular in the Asian monsoon and in the Pacific region?

## 2 Results

The strongest regional water vapour anomalies in the UTLS, as compared to the zonal mean distribution, occur in the Asian and North American monsoon circulations (Schoeberl et al., 2013), with the relative strength between Asian and American anomalies under debate (Plaza et al., 2021). At 100 hPa, a common level for studying UTLS water vapour, satellite observations by MLS (Methods) show almost 2 ppmv (around 40%) higher water vapour mixing ratios above Asia than above the Pacific

during boreal summer (Fig. 1a). Current global climate models from the CMIP6 model intercomparison project show largest regional water vapour biases in that region (Fig. 1b), and a substantial displacement of the moisture anomaly towards the Pacific when compared to satellite observations. This moisture bias has already been reported by Wang et al. (2018) and has been related to transport effects, but only for a single model. Here, we show that in the Pacific UTLS the moisture bias is a common feature of the CMIP6 models, clearly visible in the multi-model mean (Fig. 1b), and reaching even higher values for specific models (Supplement Fig. S3).

Consistent with the CMIP6 models, the EMAC climate model in its control set-up (Methods) shows a strong moisture bias in the summertime UTLS above the Pacific (Fig. 1d). In EMAC, the moist bias extends even further across the Pacific than for the CMIP6 multi-model mean, but overall lies within the spread of CMIP6 models (Supplement Fig. S3). After switching to a Lagrangian transport scheme, the modified–Lagrangian EMAC–CLaMS simulation (Methods) shows remarkably better agreement with MLS satellite observations in terms of UTLS water vapour (Fig. 1c). Hence, Lagrangian transport appears to significantly reduce the strong regional moisture bias in the summertime Pacific lower stratosphere in current global climate models. Note, that the global mean water vapour mixing ratio in the modified–Lagrangian simulation is too low compared to MLS observations, due to a known cold bias in TTL temperatures in EMAC, but the focus of the present paper is on regional anomalies.

Water vapour is a radiatively active gas, thus we expect the additional moisture above the Pacific in the control EMAC simulation to affect atmospheric temperatures and circulation. The additional moisture causes enhanced longwave cooling, which decreases temperatures in the UTLS above the Pacific. These decreased temperatures imply increased westerly zonal wind and decreased (more equatorward) meridional wind above the Pacific, consistent with thermal wind balance (Methods). Indeed, the westerlies in the Pacific UTLS close to the equator strengthen in response to the additional moisture in the Pacific UTLS (Fig. 1d, see also Discussion).

Latitude-altitude cross-sections in the Pacific UTLS show that the additional moisture in the control EMAC simulation extends well above the tropopause, causing enhanced longwave cooling and decreased temperatures particularly in the lowermost stratosphere (Fig. 2a–c). In the lowermost stratosphere, potential vorticity (PV) increases in response to the additional moisture and the isentropic PV gradient strengthens, especially around the tropopause (Fig. 2d–f). Notably, the water vapour contours in the control EMAC simulation are not following the PV structure, indicating a decoupling of the simulated water vapour distribution from transport associated with large-scale dynamics. Satellite observations, on the other hand, show a clear anti-correlation between water vapour mixing ratios and PV in the UTLS (correlation coefficient $-0.62$, slope $-0.53\,\mathrm{ppmv/PVU}$), which is opposite in the control EMAC simulation (positive correlation $0.78$ and slope $1.24\,\mathrm{ppmv/PVU}$, see Fig. 2g, h). Again, the modified–Lagrangian simulation avoids this transport bias and results in much better agreement with the observations (correlation coefficient $-0.86$, slope $-0.53\,\mathrm{ppmv/PVU}$, Fig. 2i).

The unphysical correlation between water vapour mixing ratios and PV is an indication that the moisture bias in the control simulation is caused by small-scale, unresolved processes, namely numerical diffusion in the advective transport scheme (see Charlesworth et al., 2023, and Methods). The less diffusive Lagrangian transport scheme in the modified–Lagrangian EMAC–CLaMS model results in a more efficient sampling of cold tropopause regions and in less moisture transport into the lower

stratosphere. Indeed, it is just the change in the model transport scheme and not the induced circulation changes which reduces the moist bias, as a sensitivity simulation without including the dynamical feedback of the water vapour changes provided very similar results to the modified–Lagrangian simulation with dynamical feedback included (see Supplement Fig. S1).

Moreover, the increased Pacific moisture modifies the Asian monsoon circulation. Differences in meridional velocity $v$ between the control and the modified–Lagrangian simulations (Fig. 3a) show a strengthening of the equatorward flow on the eastern edge of the monsoon anticyclone related to the water vapour induced longwave cooling, decreased temperatures and modified zonal temperature gradient $dT/dx$ above the Pacific (Methods). Thus, the increased water vapour above the Pacific in control EMAC causes a strengthened, eastward shifted equatorward flow at the eastern edge of the monsoon anticyclone, associated with a zonal broadening of the monsoon circulation.

Eastward of the Asian monsoon circulation is the region of the Pacific westerly ducts (Webster and Holton, 1982), where westerly winds may extend to and across the equator (Fig. 1e, f, white contours). Note that the term "westerly ducts" usually refers to the boreal winter circulation; here we refer to similar characteristics of the wind structure during summer. Differences between the control simulation and the modified–Lagrangian simulation of zonal $u$–wind in the subtropics (Fig. 3b) show that the westerlies above the Pacific (east of 140°E) strengthen in response to enhanced Pacific moisture, consistent with the water vapour–induced cooling and modified meridional temperature gradient $dT/dy$ (Fig. 3b, Methods). Thus, the moist bias in the Pacific UTLS is associated with strengthened westerly ducts. Slight displacements between the zonal wind and water vapour changes indicate that likely the response is not entirely local (Fig. 1 and 3).

## 3   Discussion

As the Pacific UTLS moisture bias is similar between control EMAC and CMIP6 models, also effects on regional circulation systems are expected to be similar. In the CMIP6 models, the anticyclonic Asian monsoon circulation appears displaced towards the East when compared to ERA5 geopotential height (Fig. 4a, b), in particular when considering zonal geopotential height anomaly profiles across the anticyclone center (Fig. 4c, Methods). All CMIP6 models show monsoon-associated geopotential height anomalies that are shifted eastward with respect to those in ERA5, with related differences larger than year-to-year variability (Fig. 4c, red error bars).

Based on similarity between CMIP6 inter-model correlations with the effects of the Pacific moist bias in the EMAC model experiment, we hypothesize in the following that CMIP6 model differences could partly be related to the moisture content in the Pacific UTLS. The CMIP6 inter-model correlation between local meridional $v$–wind velocity in the monsoon anticyclone and a water vapour index measuring the strength of the Pacific moisture anomaly (Methods) shows a significant anticorrelation above about 100 hPa and eastward of 120°E (Fig. 3c, with the mean correlation for winds averaged over a broader region shown in Fig. 3e). In other words, those models with a moister lower stratosphere above the Pacific simulate an anticyclonic monsoon circulation which is displaced further east. Above the tropopause in the lower stratosphere, the pattern of negative/positive CMIP6 inter-model correlation resembles the pattern of negative/positive differences in meridional flow in the EMAC model experiment (Fig. 3a). Hence, the enhanced equatorward flow at the eastern monsoon anticyclone edge appears related to the

strength of the water vapour anomaly above the Pacific, in both the EMAC model experiment and in CMIP6 models. Those models with a moister Pacific UTLS simulate an eastward displaced monsoon anticyclone.

Prevailing westerlies in the Pacific UTLS, the westerly ducts, allow Rossby waves to propagate further towards, even across the equator (Waugh and Polvani, 2000), and to cause inter-hemispheric exchange (Yan et al., 2021). Therefore, a realistic representation of the westerly ducts in atmospheric models is important for correctly simulating the global distribution of trace gases and pollutants, especially for species with strong emissions in the Northern hemisphere. Zonal profiles of zonal wind in the northern tropics show that most CMIP6 models simulate too strong westerlies above the Pacific (Fig. 4d) compared to ERA5. Furthermore, CMIP6 inter-model correlations between zonal wind and Pacific UTLS moisture show a significant correlation in the Pacific subtropics around the tropopause (Fig. 3d, f), although weaker than for meridional wind at the eastern monsoon edge (Fig. 3c). The correlation for zonal wind implies that those models with increased Pacific moisture simulate stronger westerly ducts, consistent with the effect of enhanced Pacific moisture in the EMAC model experiment (Fig. 3b). Hence, the overestimated westerly ducts in CMIP6 models appear to be related, at least partly, to the model moist bias above the Pacific.

Differences in the simulated moisture bias are likely related to differences in diffusive model transport which are caused by differences in resolution or in the transport scheme. Across the CMIP6 ensemble, the strength of the bias is not related in a simple manner to horizontal or vertical model resolution. Although the smallest biases occur in those models with relatively high vertical resolution, different models with relatively high number of levels or horizontal grid points simulate very different biases (Fig. 3e–f). Exploring these issues in more detail would require dedicated model experiments with changes in the resolution in the UTLS.

Differences in model transport could also affect other trace gas species with large gradients, like ozone. However, as water vapour shows particularly steep gradients in the UTLS also the associated effects on circulation are expected to be comparably strong.

## 4   Conclusions

We show that a distinct moist bias above the Pacific in the summertime UTLS is a common feature of current climate models. This regional moist bias is not related to large-scale transport characteristics and the expected anti-correlation between UTLS water vapour and PV breaks down. The bias can be significantly reduced by applying a Lagrangian, numerically less diffusive transport scheme, as realized in the modified–Lagrangian EMAC–CLaMS simulation. The analysis of a dedicated climate model experiment and CMIP6 models further shows that the Pacific moist bias affects regional circulation patterns by enhancing local longwave cooling and modifying temperature gradients. In particular, our results indicate that the water vapour increase in the Pacific UTLS strengthens the Pacific westerly ducts and causes zonal displacement of the Asian monsoon upper level anticyclonic flow. Hence, improving the representation of water vapour in the Pacific UTLS region in models, for instance by employing a less diffusive (Lagrangian) transport scheme, will affect regional circulation systems and could be a promising

way to improve model simulations of current and future climate conditions, for instance in the Asian monsoon and Pacific regions.

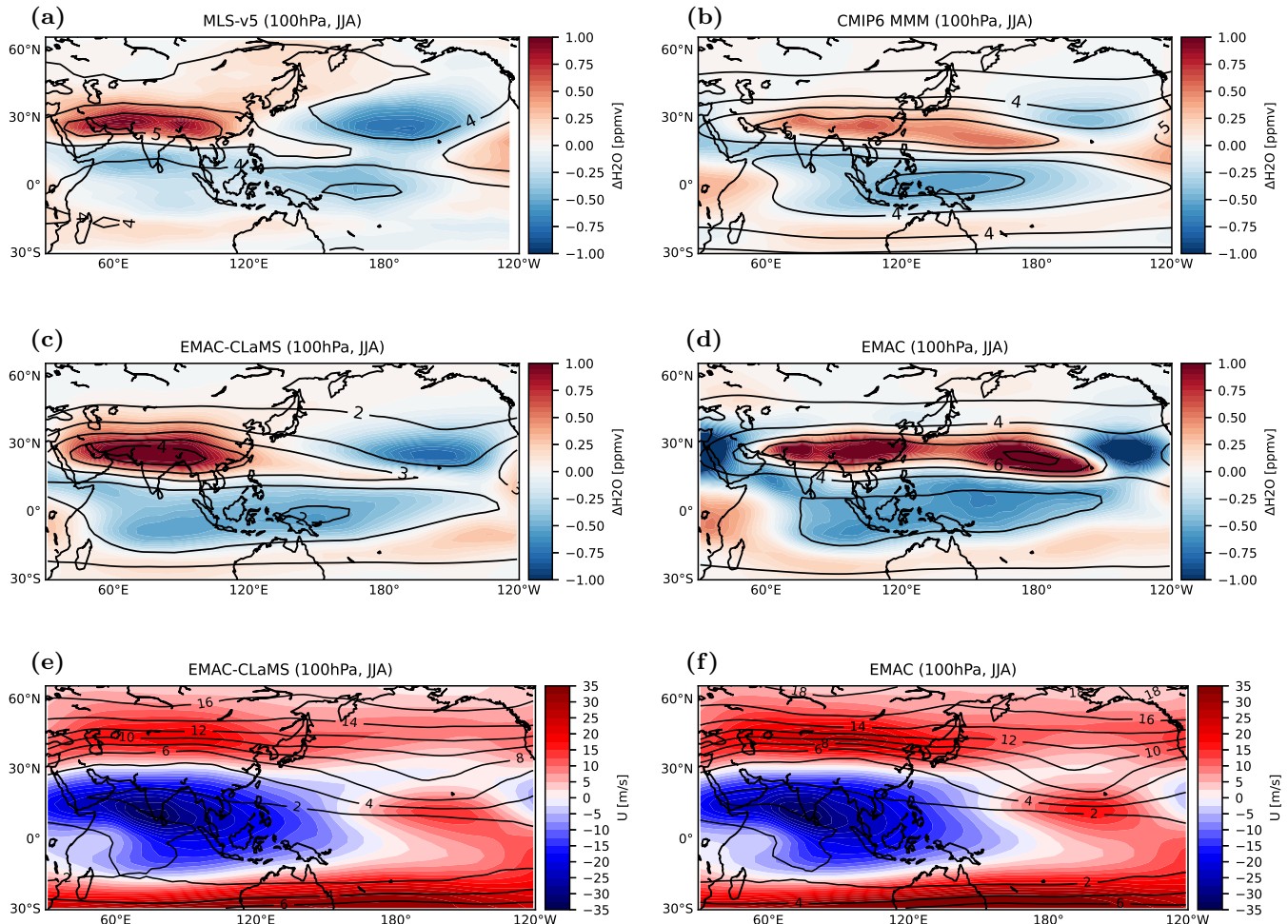

**Figure 1.** Water vapour zonal anomalies at 100 hPa in boreal summer (June–August) from (a) MLS satellite observations and from different climate model simulations, with (b) showing CMIP6 multi-model mean, (c) modified–Lagrangian EMAC–CLaMS and (d) control EMAC. Black contours show climatological water vapour mixing ratios (1 ppmv steps). (e–f) Zonal wind velocity at 100 hPa from EMAC–CLaMS (left) and EMAC (right), together with the climatological PV distribution (black contours, 2 PVU steps).

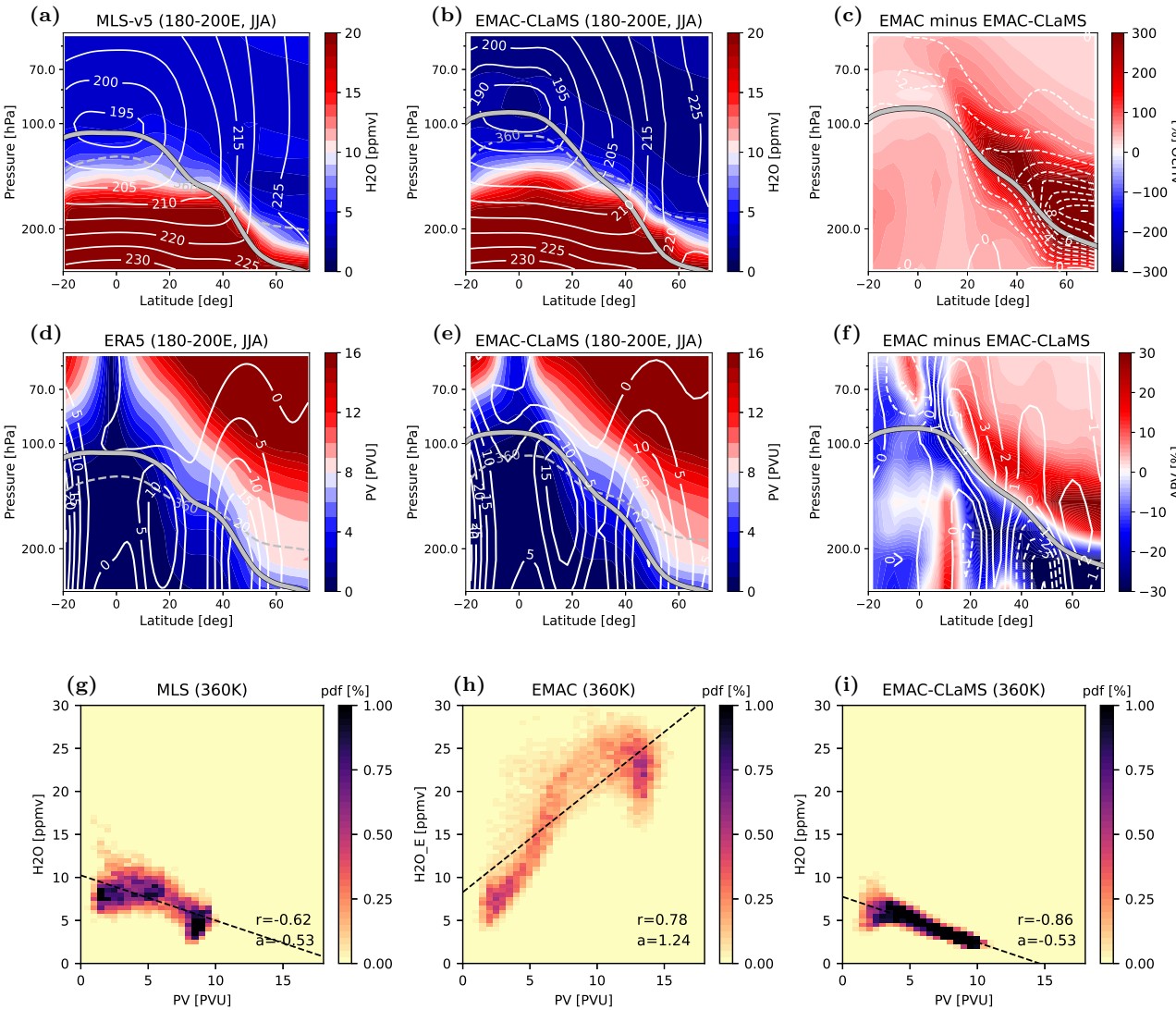

**Figure 2.** Water vapour cross-sections in the Pacific region (180°-200°E) in boreal summer (June–August), for (a) MLS satellite observations, (b) modified–Lagrangian EMAC–CLaMS climate model simulation, and (c) the difference between control EMAC and Lagrangian modified EMAC–CLaMS simulations. White contours show temperature (a, b) and the temperature difference between the two simulations (c). The thick gray line is the WMO lapse rate tropopause, the grey dashed line the 360K isentrope. (d–f) Same but for PV color coded, and zonal wind as white contours (in d, PV is taken from ERA5). (g–i) Correlation between water vapour and potential vorticity at 360 K potential temperature level in the Pacific region (15°-70°N, 140°-240°E) for (g) MLS satellite observations and ERA5 reanalysis PV, (h) control EMAC, and (i) modified–Lagrangian EMAC–CLaMS. The Pearson correlation coefficient $r$ and linear regression slope $a$ are given in each figure and the linear regression fit is illustrated as black dashed line.

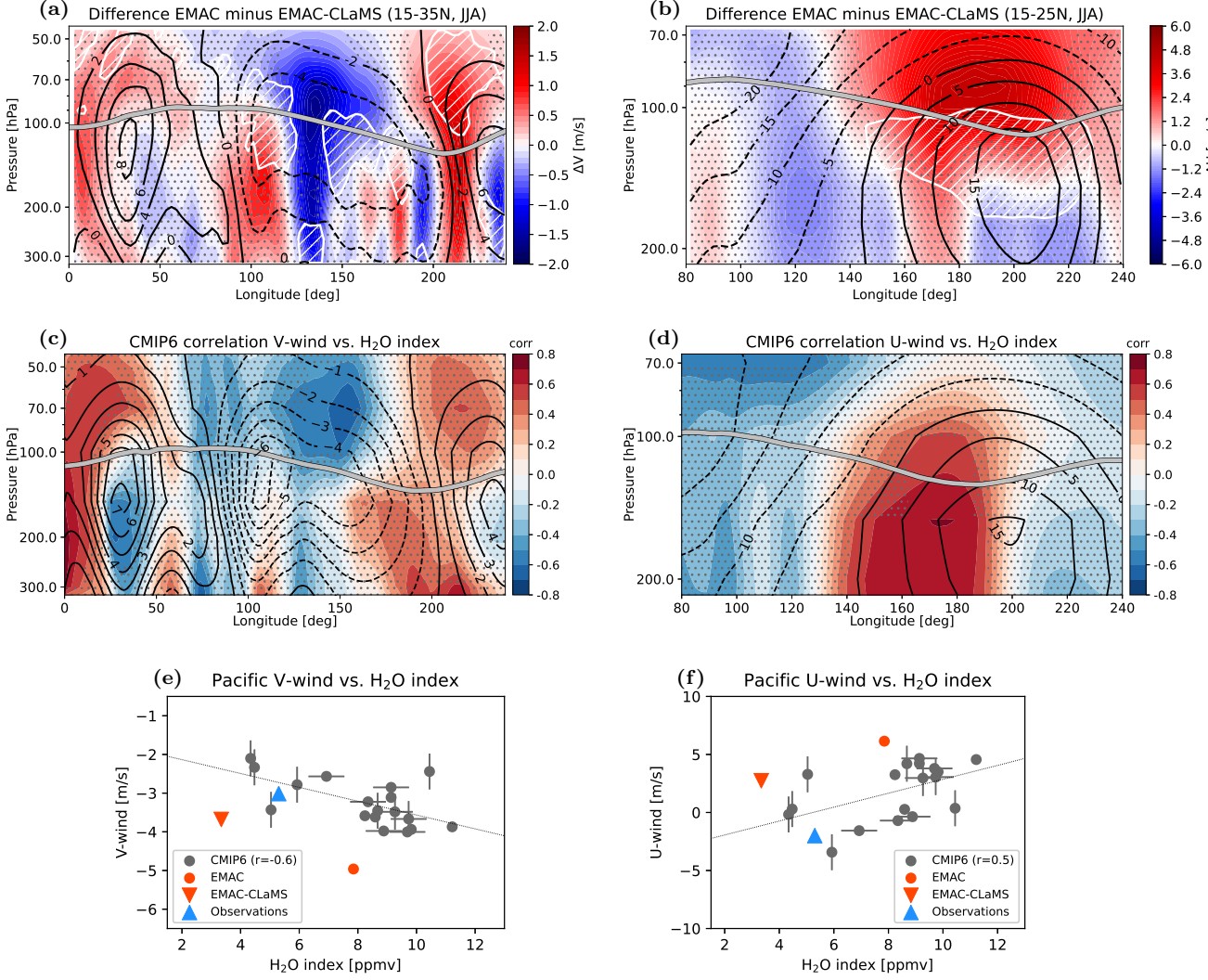

**Figure 3.** (a) Difference in meridional $v$–wind between control EMAC minus modified–Lagrangian EMAC–CLaMS for a longitude section across the Asian monsoon ($15° - 35°$N, June–August). Black contours show climatological $v$ (from Lagrangian modified–EMAC simulation; northward solid, southward dashed), the thick grey line is the tropopause. White hatching highlights regions of negative model difference of zonal temperature gradient ($dT/d\lambda < -0.015$ K/deg, $\lambda$ longitude). Black dots indicate where the difference is not statistically significant at 95% confidence level. (b) Same model difference, but for zonal $u$–wind, and as longitude section across the Pacific ($15° - 25°$N). Black contours show climatological zonal wind. White hatching highlights regions of negative model difference of meridional temperature gradient ($dT/d\phi < -0.08$ K/deg, $\phi$ latitude). (c) The CMIP6 inter-model correlation between meridional $v$ and the Pacific $H_2O$ index measuring the strength of the Pacific moisture anomaly across models (see Methods), for the same longitude section as in (a). Solid black lines show multi-model mean meridional wind, the grey line the tropopause from ERA5. Black dots indicate where the correlation is not significant at 95% confidence level. (d) Same inter-model correlation, but for zonal wind and the same longitude section as in (b). Solid black lines show multi-model mean zonal wind. (e) Inter-model correlation between Pacific $H_2O$ index and $v$–wind averaged in the Pacific UTLS ($15° - 35°$N, $130° - 155°$E, $100 - 70$ hPa). Vertical lines indicate CMIP6 models with relatively high vertical resolution, horizontal lines those with high horizontal resolution, "observations" corresponds to MLS $H_2O$ and ERA5 wind. The correlation coefficient for CMIP6 models is given in the legend. (f) Same, but for the correlation with zonal $u$-wind averaged over $15° - 25°$N, $160° - 185°$E, $130 - 90$ hPa.

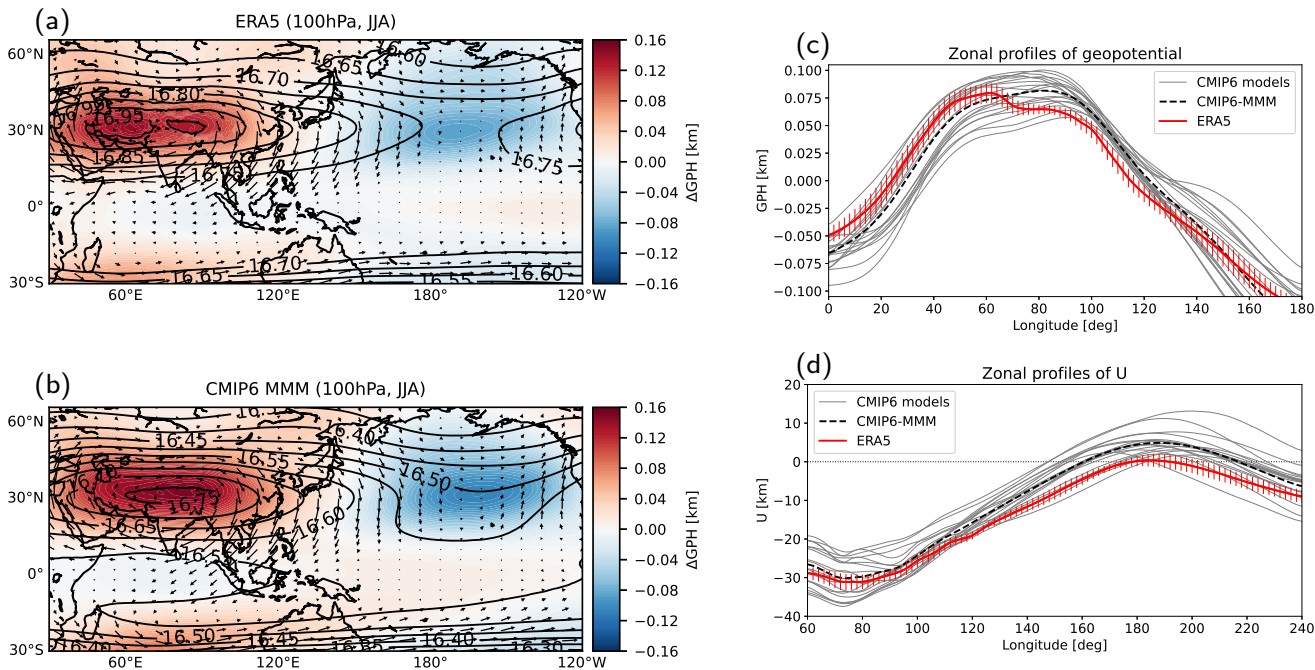

**Figure 4.** Geopotential height zonal anomalies at $100\,hPa$ for boreal summer (June–August), from (a) ERA5 and (b) CMIP6 multi-model mean. Black contours show climatological geopotential height and black arrows illustrate horizontal wind velocity. (c) Zonal profiles of geopotential height at the latitude of the monsoon geopotential height maximum (Methods), from CMIP6 models (grey), multi-model mean (black dashed) and ERA5 (red). Red error bars show the standard deviation of year-to-year variability for ERA5 data. (d) Zonal profiles of zonal wind across the Pacific at $15°N$ from CMIP6 models (grey), multi-model mean (black) and ERA5 (red).

 ## 5   Appendix

### 5.1   Methods

#### 5.1.1   Satellite observations

The simulated water vapour distributions from the different models are compared to observations from the Microwave Limb
Sounder (MLS) instrument onboard the Aura satellite (version 5 data), which has started operation in 2004. MLS provides
 a relatively high sampling of the globe from about 82°S-82°N, with about 3500 profiles per day. In the UTLS region, the
MLS water vapour product has a vertical resolution of about 3 km. The MLS averaging kernels have been shown to induce
artifacts in the water vapour distribution, in particular vertical oscillations at high latitudes (Ploeger et al., 2013), but have no
large influence on lower, subtropical latitudes, which are the focus of this paper. As the main goal of the paper is to investigate
differences between model simulations, we refrain from smoothing the model profiles with the satellite averaging kernels. For
 comparison with the model data, monthly mean climatologies have been compiled from the MLS data on the original MLS
pressure levels and for the period 2005–2015. For further information on MLS water vapour and the retrieval technique see
Read et al. (2007).

#### 5.1.2   EMAC model simulations

The chemistry climate model used for this study is the ECHAM/MESSy Atmospheric Chemistry model (EMAC). EMAC
 couples the ECHAM5 dynamical core to physical processes via the Modular Earth Submodel System (MESSy) middleware
(for details see Jöckel et al., 2016, and references therein). The simulations analyzed for this study employ a T42 spectral
resolution (about $2.8 \times 2.8$ degrees latitude-longitude resolution), 90 vertical levels from the surface to 0.01 hPa, and are free-
running with prescribed sea surface temperatures and radiatively active substances (for further details see Charlesworth et al.,
2023). The simulation period spans 40 years starting 1970, with only the last ten years (2000–2009) considered for this paper.
 We ensured that the presented results are insensitive to the choice of period by checking robustness for the period 1990–1999
(and for 1979–2020 for ERA5 variables).

   Water vapour is calculated online and includes methane oxidation as chemical source in the stratosphere. In particular, we
compare two different simulations with stratospheric water vapour transported either with the Flux-Form Semi-Lagrangian
standard EMAC transport scheme (denoted "control" EMAC) or with the Lagrangian transport scheme CLaMS (Chemical
 Lagrangian Model of the Stratosphere), denoted "modified–Lagrangian" EMAC, as described by Charlesworth et al. (2023).
Due to its Lagrangian nature, the CLaMS transport scheme has largely reduced numerical diffusion compared to the standard
EMAC transport (Hoppe et al., 2014; Charlesworth et al., 2020). In both simulations the water vapour is coupled to the model
radiation and thereby affects atmospheric temperature and circulation. Note that the Lagrangian water vapour calculation for
this study only ranges from about 250 hPa (more precisely the model level closest to 250 hPa) to the model top, to ensure that
 model differences originate solely from stratospheric water vapour. As the only difference between both simulations is the used

transport scheme for stratospheric water vapour, differences in temperatures and circulation can be unambiguously attributed to the transport scheme.

To further clarify the roles of the changed transport scheme versus the induced dynamical effects for the water vapour model differences, an additional sensitivity simulation was carried out with stratospheric water vapour calculated with Lagrangian CLaMS transport but not coupled to radiation. The close similarity of the lower stratospheric water vapour distribution with the modified Lagrangian simulation (with stratospheric water vapour coupled to radiation) shows that the water vapour differences compared to the control EMAC simulation are mainly related to the difference in the transport scheme (numerical diffusion) and not to the induced dynamical effects (Supplement Fig. S1).

A second sensitivity simulation was carried out for the first year of the simulation period with tendency output for the control EMAC water vapour. These additional diagnostic terms which are output by EMAC include the tendencies due to advection, clouds (mainly dehydration processes at levels around the tropopause), convection, and parameterized vertical diffusion (Supplement Fig. S2). Consideration of the different model tendencies in the EMAC model shows that the moistening in the Pacific lower stratosphere in control EMAC is the net effect of the interplay of advection of moist tropospheric air masses and dehydration by cloud processes (Supplement Fig. S2). In the region of the moist bias over the Pacific, advective moistening dominates and causes a net moist bias there during boreal summer. Hence, the excessive moisture transport in the standard EMAC model transport scheme appears related to excessive numerical diffusion in the advective transport. This EMAC sensitivity analysis does not exclude a more significant effect of convection in other models or in the real atmosphere.

### 5.1.3 ERA5 reanalysis

The ERA5 reanalysis from the European Centre for Medium-Range Weather Forecasts (ECMWF) covers the period from 1949 onwards (Hersbach et al., 2020). Here, we use monthly mean climatologies compiled from ERA5 wind, PV, geopotential height and temperature data for the period 2005–2015, for better comparability with MLS observations and the climate model simulations. The ERA5 data assimilation system is based on the ECMWF Integrated Forecast System, cycle CY41R2, and uses a 4D-Var assimilation scheme. ERA5 data is available hourly, with a horizontal resolution of about $30\,\mathrm{km}$ (T639 spectral resolution), and a vertical range from the surface to about $0.01\,\mathrm{hPa}$ (137 hybrid levels). For the monthly mean climatologies shown in this paper we used ERA5 data every 6 hours, truncated to a $1° \times 1°$ latitude-longitude grid and with full vertical resolution, as provided by ECMWF.

### 5.1.4 CMIP6 model intercomparison project

To place the EMAC simulations into context, we compare to climate model simulations from the Coupled Model Intercomparison Project, Phase 6 (CMIP6). CMIP6 is a multi-model intercompison activity which has been carried out in support of the Sixth Assessment of the IPCC (AR6). Here, we consider the historical simulations which are fully coupled model simulations, with external forcings from solar variability, volcanic aerosols, and anthropogenic emissions (greenhouse gases, aerosol) following observations. If ozone chemistry is not included, the models use prescribed time-varying ozone concentrations. These simulations cover the period 1850–2014, but we only use the data for 2000–2014 for better comparability with the other data

sets. From the CMIP6 models investigated recently by Keeble et al. (2021) and which have been shown to have reasonable stratospheric water vapour, those 18 models with the necessary data for this study are: AWI-ESM-1-1-LR, BCC-CSM2-MR, BCC-ESM1, CESM2, CESM2-FV2, CESM2-WACCM, CESM2-WACCM-FV2, CNRM-CM6-1, CNRM-ESM2-1, E3SM-1-1, GFDL-CM4, IPSL-CM6A-LR, MPI-ESM-1-2-HAM, MPI-ESM1-2-HR, MPI-ESM1-2-LR, MRI-ESM2-0, NorESM2-MM, SAM0-UNICON (for further details on the water vapour in these models see Keeble et al., 2021).

### 5.1.5 Inter-model correlations

The circulation response to increased stratospheric water vapour above the Pacific in CMIP6 models is analyzed statistically by using inter-model correlations. For that purpose, a Pacific $H_2O$ index is defined by averaging lowermost stratospheric water vapour in the Pacific region (140–220°E longitude, 20–45°N latitude, 150–70 hPa pressure) in each model. This index measures the strength of the Pacific moisture anomaly in different climate models and is correlated with circulation variables (zonal and meridional wind) across models. Hence, a negative correlation with meridional $v$-wind in the lower stratosphere above the Pacific (about 100–160°E) in Fig. 3, for instance, means that those models with higher water vapour mixing ratios in the Pacific UTLS simulate more negative, equatorward meridional wind in this region. Consequently, the eastern, equatorward flank of the monsoon anticyclonic circulation is strengthened in these models. Similar inter-model correlations have been used for analyzing the westerly ducts in Fig. 3.

In addition, CMIP6 models are classified with respect to their horizontal and vertical resolution in Fig. 3 e–f (with the information on model resolution taken from Keeble et al., 2021). For that reason, those models with more than 70 vertical levels are classified as models with relatively high vertical resolution (MPI-ESM1-2-HR, CESM2-WACCM, CESM2-WACCM-FV2, CNRM-CM6-1, CNRM-ESM2-1, E3SM-1-1, IPSL-CM6A-LR, MRI-ESM2-0), while those models with more than 150/280 longitude/latitude grid points are classified as models with relatively high horizontal resolution (MPI-ESM1-2-HR, CESM2-WACCM, CESM2, BCC-ESM1, GFDL-CM4, NorESM2-MM, SAM0-UNICON).

However, it is not necessarily the total number of levels or horizontal grid points but more likely the resolution in the tropopause region which matters for stratospheric water vapour. Therefore, the model classification discussed above only provides an indication that the model characteristics and processes involved in causing the bias are complex. For a thorough analysis of model resolution and diffusion effects additional sensitivity simulations with a particular model but employing different horizontal and vertical resolutions as well as different transport schemes would be needed.

### 5.1.6 Dynamical balances

The large-scale effects on atmospheric circulation from changes in stratospheric water vapour can be understood from simplified, balanced dynamics (Charlesworth et al., 2023), taking into account the radiative effect of water vapour on atmospheric temperatures. An increase of water vapour in the lowermost stratosphere causes local long-wave cooling and thereby modifies the atmospheric temperature gradients in that region. These changes in meridional and zonal temperature gradients $\Delta\left(\partial_y T\right)$

and $\Delta\left(\partial_x T\right)$ are related via the thermal wind relation to changes in $u$ and $v$–wind velocities

$$\partial_z(\Delta u) = -\frac{R}{H f}\Delta\left(\partial_y T\right)\ ,\qquad \partial_z(\Delta v) = \frac{R}{H f}\Delta\left(\partial_x T\right)\ . \tag{1}$$

For instance, the decrease in the meridional temperature gradient below about $90\,\mathrm{hPa}$ above the Pacific (around $200°\mathrm{E}$) in Fig. 3b is related to an increase in the vertical gradient of zonal wind. Likewise, the increase in the meridional temperature gradient above is related to a decrease in the vertical zonal wind gradient. Hence, the increased water vapour in the Pacific lowermost

stratosphere causes increased westerlies in that region, and thereby strengthens the Pacific westerly ducts. Similar relations hold between the changes in the zonal temperature gradient and changes in the vertical gradient of meridional wind (see Eq. 1) such that negative meridional $v$-wind velocities further decrease around and above the tropopause in the Pacific (at about $130°\mathrm{E}$), implying a strengthened equatorward flow (Fig. 3). Consequently, increased lower stratospheric water vapour above the Pacific is related to a strengthened equatorward flank on the eastern side of the monsoon anticyclone.

### 5.1.7 Monsoon anticyclone diagnosis

Different diagnostics have been proposed in the literature for assessing strength and extend of the Asian monsoon anticyclone, based on either geopotential height, Montgomery stream function of potential vorticity (Randel and Park, 2006; Ploeger et al., 2015; Santee et al., 2017). Due to data availability for the considered model simulations we here use geopotential height anomaly as a measure for the anticyclonic circulation. In particular, we follow Bergman et al. (2013) and diagnose the center

of the anticyclone as the location of maximum geopotential height on the $100\,\mathrm{hPa}$ isobaric surface. Longitude profiles at the latitude of the anticyclone center are then used to compare the longitudinal extent of the upper level anticyclonic monsoon circulation in different models (Fig. 4).

*Code and data availability.* The EMAC and EMAC–CLaMS models are available in the Modular Earth Submodel System (MESSy) git database. Detailed information is available at https://messy-interface.org/licence/application. ERA5 reanalysis data are available from the

275 European Centre for Medium-range Weather Forecasts (https://apps.ecmwf.int/data-catalogues/era5/?class=ea, last access: 30 November 2023). The EMAC and EMAC–CLaMS model data used for this paper may be requested from the corresponding author (f.ploeger@fz-juelich.de). CMIP6 model data is publically available at https://esgf-node.llnl.gov/search/cmip6.

*Author contributions.* FP initiated the study, carried out the analysis and wrote the manuscript. EC set-up and carried out the model simulations and the data processing. TB and PK were strongly involved in several detailed discussions during different phases of the project. All

280 authors contributed to writing the manuscript.

*Competing interests.* One of the co-authors (RM) is a member of the editorial board of Atmospheric Chemistry and Physics. The peer-review process was guided by an independent editor, and the authors have also no other competing interests to declare.

*Acknowledgements.* We thank Nicole Thomas and Patrick Jöckel for support with setting up the model simulations. This study was funded the Deutsche Forschungsgemeinschaft (DFG, German Research Foundation) – TRR 301 – Project-ID 428312742. Finally, we gratefully ac-

knowledge the computing time for the CLaMS simulations which was granted on the supercomputer JURECA at the Jülich Supercomputing Centre (JSC) under the VSR project ID CLAMS–ESM.

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
