# Peer review of "Moist bias in the Pacific upper troposphere and lower stratosphere (UTLS) in climate models affects regional circulation patterns"

_EGUsphere, 2023_

## Author Comment (AC1)

**Reply to Reviewer 1**

We thank the Reviewer for the careful reading and evaluation of the manuscript and the good comments which helped a lot to further improve the paper. In the following, we address all comments and questions raised (Reviewer's comments in italics). Text changes in the manuscript are highlighted in color (except minor wording changes).

Besides several specific comments, we see two common main concerns raised by both Reviewer's, regarding (i) the presentation quality and clarity of figures, and (ii) the discussion of specific climate model characteristics. A short overview of the related changes in the revised manuscript is:

(i) To enhance the **presentation quality and clarity** we modified all figures in the revised manuscript. In particular, we chose different color schemes, reduced the number of contours, changed to difference plots for certain quantities and improved the model-correlation visualization (a detailed list of figure changes is in the reply to the specific comment below). We are confident that these changes clearly improve the presentation of our results.

(ii) To relate our results to specific **model characteristics**, we included a new subfigure Fig. 3 (e, f). This figure shows the inter-model correlation for wind velocities averaged over the region of maximum signal, and includes information on horizontal and vertical resolution of the models by highlighting models with relatively high number of horizontal grid points with horizontal lines and models with high number of levels with vertical lines. Based on this simple model classification, no clear relation of resolution differences to the simulated water vapour bias is found. These findings are briefly discussed at the end of the discussion section and described in more detail in the Methods section 5.1.5.

A more detailed reply to all comments and description of the changes in the revised manuscript is given below.

**General comment:**
*This is an interesting work finding an interpretation and a remedy to one of the serious known limitations of climate models. The manuscript is well written and I have only a limited number of minor comments and questions to be considered by the authors.*
Thanks for this positive evaluation of the manuscript!

**Specific comments:**

*Solomon et al. (2010) wrote that variations of lower stratospheric water vapour may account for up to 30% of the greenhouse gases modification of the radiative budget on a decadal scale. They do not write that they account for 30% of the total variation since 1850 as suggested by the sentence on lines 13-14 in the manuscript. Please correct to avoid such confusion.*
Thanks for pointing this misleading formulation out (as similarly by the other Reviewer)! We changed the sentence to: "Decadal variations in stratospheric water vapour have been shown to modify the radiative budget on a decadal scale by up to 30%."

*It is quite clear that the excessive numerical diffusivity of the CMIP6 models should be related to their spatial resolution and their transport scheme, which display large differences among the ensemble. It is quite frustrating that they are here all put in the same bag without any attempt to draw a distinction. For instance, it would be very interesting to know whether it is the horizontal rather than the vertical resolution that matters as very different choices have been made among the ensemble. In terms of horizontal resolution, the T42 resolution f the EMAC used here put is at the lower end of the CMIP6 ensemble but its vertical resolution puts it at the upper end. This is perhaps an answer to the above question as fig.1 shows it does much worse than the CMIP6 mean although what probably matters is not the total number of levels but the number of those which span the UTLS, a parameter which is badly documented.*
We totally agree with the Reviewer that it would be beneficial to know whether it is the vertical or horizontal model resolution that matters more in causing the water vapor bias, and that the resolution in the UTLS matters most. Unfortunately, this question is not easy to answer with the available suite of models and simulations. The water vapor maps for single models in the supplement (Fig. S3) show that indeed the model with highest resolution (MPI-ESM1-2-HR) simulates a well confined monsoon moisture anomaly and only a weak Pacific moisture bias. However, the differences in water vapor bias to other models are not related in a simple manner to differences in vertical or horizontal resolution. For instance, models with either a comparable number of vertical levels (e.g., MRI-ESM2-0) or with a comparable number of latitude/longitude grid points (e.g., BCC-CSM2-MR) show stronger moisture biases than the model with highest resolution (MPI-ESM1-2-HR). Furthermore, as already said by the Reviewer, regarding vertical resolution it is likely the number of levels in

the UTLS rather than the total number of levels, and information on that is not well documented. In general, as the transport schemes employed by the different models are quiet different, the differences in the simulated water vapor distribution are not simply explaniable by differences in the number of vertical levels or horizontal grid points.

To discuss these aspects in the revised manuscript, we included a new subfigure Fig. 3 (e–f) which shows the inter-model correlation between Pacific $u$- and $v$-wind with the Pacific water vapor index for the winds averaged over the UTLS region of most significant correlation (see figure caption and Methods). In addition we selected from the CMIP6 models a few models with relatively high horizontal and/or vertical resolution (indicated by the horizontal/vertical lines in the scatter plot). The criteria for selecting these models are given in the Methods section 5.1.4. Clearly, neither differences in horizontal nor in vertical resolution explain the spread of simulated Pacific moisture bias. Hence, a more detailed analysis of model differences, best including sensitivity simulations with the same model and differing horizontal/vertical resolution would be needed to draw robust conclusions on this issue. We discuss these points in the revised manuscript in a short paragraph at the end of the discussion section and in the Methods section 5.1.5.

*My main concern is about the figures and their readability. Black contours and labels on dark blue and red are hardly visible and readable. This is not good on the screen and it is terrible on a printed version. This needs to be improved. There is no reason to use a divergent color map to show water vapour in fig2 (a-c). The two first rows of fig.2 show redondantly PV and U on panels which are overcrowded. Please reorganise these two rows to show only 3 variables in each panel. In figs 1 and 2, some quantities (wind, temperature) would be better displayed as differences between EMAC-ClaMS and EMAC. Adding a grid would overload the figures but ticks can be put on the upper and right sides of the figures to improve readability. In figure 3, I do not think that the temperature gradient contours are very useful, and they have no labels and there is no indication of contour intervals in the caption. I would prefer to have some contours for the quantities displayed in color as it is almost impossible to read the values from the color map (or choose a better indexed color map).*

Thanks for this very helpful comment, which was similarly brought up also by the other Reviewer! We agree that the clarity and readability of many figures should indeed be improved. For the revised manuscript we did significant efforts to improve the presentation quality. The changes to the figures are summarized below:

- We reduced the number of contours plotted by removing redundancy and focussing only on the most relevant variables. Therefore, PV contours are not shown in Fig. 1 a–d anymore, but only in Fig. 1 e–f. Here, we now show PV together with the zonal wind distribution at 100 hPa in the two EMAC model simulations to illustrate the circulation differences more clearly. Figure 2 is even more substantially reorganized, by showing the observation and Lagrangian EMAC–CLaMS model water vapor and PV fields together with the model difference between the two EMAC simulations (Fig. 2 c, f). The water vapor plots also show temperature contours (and only these), while the PV plots now show zonal wind contours (model differences in Fig. 2 c, f). With this reorganization of figures still the relevant variables are shown, but with much more clarity.

- The color scheme for anomalies and differences is changed to schemes with somewhat lighter maximum blue and red colors, so that dark contours are still readable. Also contour colors have been changed to have less different colors and still ensure optimal contrast (e.g. only black contours in Fig. 1, only white contours in Fig. 2, tropopause as grey lines in Figs. 2 and 3).

- Circulation differences between the two EMAC model simulations (control vs. modified Lagrangian) are now shown in difference plots in Fig. 2 c, f. These new subfigures show the temperature, zonal wind and PV differences as induced by the stratospheric water vapor differences with much more more clarity compared to the manuscript version before. In particular, the cooling effect as well as the strengthening of PV gradients of the enhanced lowermost stratospheric water vapor can be more clearly seen now.

- Regions of relevant temperature gradient changes explaining the wind changes in Fig. 3 a, c are now highlighted with white hatching. Although not including quantitative information on the exact values anymore, the modified figure still shows qualitatively that the relation between vertical wind shear and horizontal temperature gradient changes is consistent with thermal wind balance.

We think that with these changes the presentation of our results has become much clearer.

*It is very hard to appreciate from the first two rows of fig. 2 that the isentropic PV gradient is strengthened around the tropopopause and whether vapour contours are following or not the PV structure, although it is quite clear from the third row.*

In the revised version, the strengthened PV gradient becomes much clearer from the difference plot in the new

Fig. 2f. The same holds for the relation between water vapor and PV changes around the tropopause. (See also the reply to the comment before regarding the presentation quality of figures).

*Figure 3 shows that the zonal wind incease it shifted by 30°E with respect to the water vapour anomaly. So the temperature drop should also be shifted which means that the response to the water vapour is not the simple local process advocated in the manuscript but involves also transport and delay.*
Thanks for pointing to that. A short related remark is added at the end of the results section Sect. 3: "Slight displacements between the zonal wind and water vapour changes indicate that likely the response is not entirely local (Fig. 1 and 3)."

*It should be noted that the differences between EMAC and EMAC-CLaMs are much smaller than those between ERA5 and EMAC except perhaps for the PV in the lower stratosphere. In this respect it would be useful to see the curves for EMAC and EMAC-ClaMS in fig.4 (c-d) to appreciate the improvement in regard to the current dispersion and bias of the models.*
As the model configuration for the EMAC simulations used here is not exactly the same as for the CMIP6 simulations, we refrain from including a too close comparison. Some comparison between the water vapour and circulation biases between these models can be seen in the new Fig. 3 e–f, but due to the above differences in model set-up we don't discuss these more thoroughly.

*The manuscript is not totally clear about the effect on the monsoon circulation. It is indicated that the equatorward branch is broadened and strengthened on the eastern side but this is not a mechanism which by itself is able to modify the closed monsoon circulation as dicussed in section 3 since this modification is correlated to its internal PV budget.*
To understand the complete non-local circulation response the application of a mechanistic model would be preferred. This is, however, beyond the scope of the present paper but could be a good, important focus of future work. Moreover, the main effect of the increased water vapour in the Pacific UTLS on the monsoon anticyclone is an eastward displacement associated with the enhanced longwave cooling over the Pacific. We are more careful with the related formulations in the revised manuscript (e.g. in the second paragraph of the discussion).

*I am unsure the proper way to refer to ERA5 data is a link to Lawrence Livermore National Laboratory.*
There was an error in the cited data link for ERA5, which is corrected in the revised version. Thanks for pointing this out!

---

## Author Comment (AC2)

**Reply to Reviewer 3**

We thank the Reviewer for the careful reading and evaluation of the manuscript and the good comments which helped a lot to further improve the paper. In the following, we address all comments and questions raised (Reviewer's comments in italics). Text changes in the manuscript are highlighted in color (except minor wording changes).

Besides several specific comments, we see two common main concerns raised by both Reviewer's, regarding (i) the presentation quality and clarity of figures, and (ii) the discussion of specific climate model characteristics. A short overview of the related changes in the revised manuscript is:

(i) To enhance the **presentation quality and clarity** we modified all figures in the revised manuscript. In particular, we chose different color schemes, reduced the number of contours, changed to difference plots for certain quantities and improved the model-correlation visualization (a detailed list of figure changes is in the reply to the specific comment below). We are confident that these changes clearly improve the presentation of our results.

(ii) To relate our results to specific **model characteristics**, we included a new subfigure Fig. 3 (e, f). This figure shows the inter-model correlation for wind velocities averaged over the region of maximum signal, and includes information on horizontal and vertical resolution of the models by highlighting models with relatively high number of horizontal grid points with horizontal lines and models with high number of levels with vertical lines. Based on this simple model classification, no clear relation of resolution differences to the simulated water vapour bias is found. These findings are briefly discussed at the end of the discussion section and described in more detail in the Methods section 5.1.5.

A more detailed reply to all comments and description of the changes in the revised manuscript is given below.

**Overall comment:**

*Ploeger et al. report on the impact of moisture biases in the Pacific UTLS on regional circulation patterns at the tropopause level, such as the anticyclone Monsoon circulation and its zonal extent over the Pacific. They show that a modified Lagrangian scheme improves a ubiquitous deficiency of the EMAC model, in terms of the UTLS climatology of water vapor (too "zonal" and moist bias over W-Pacific) and this improvement also has effects on the circulation, such as on the Monsoon anticyclone structure (reducing the zonally too broad and strong Monsoon anticyclone, and too strong westerlies over the tropical Pacific). I find this study really interesting, and the paper is well structured. I only have some minor suggestions. I recommend prompt publication upon addressing them.*
Thanks for this positive evaluation of the manuscript!

**General comments:**

*1) I think that for CLAMS, the role of stratospheric water vapor (SWV) can be indeed nicely isolated with the suite of experiments presented in the paper. However, I am less convinced about the "generalization" for all CMIP6 models: I am not entirely sure that the "cross correlation" of U-wind vs SWV (Fig. 4) can be really taken as "proof" that moist biases in other models have the same effects on the circulation as demonstrated for EMAC. This is essentially shown in Fig. 4c and 4d. I see quite a difference, for example, between the effects of CLAMS in EMAC (panel b - this should be the "impact of SWV improvements") and the relationship between inter-model spread in SWV and U in CMIP6 (panel d) - the location of these correlations is quite different. Can the authors maybe test the correlation between SWV and U (on e.g. inter-annual time-scales instead of "inter model") within CLAMS and EMAC directly, to support their inferences about dynamical impacts of SWV biases in other climate models?*
We do fully agree with the reviewer that the CMIP inter-model correlation alone should not be taken as a "proof" for the effects of moist biases on circulation. In the EMAC model experiment, on the other hand, the effect of stratospheric water vapour changes can be well isolated, and the EMAC control minus modified–Lagrangian differences (e.g. Fig. 3 a/b) unambigously show the effects of stratospheric water vapour differences in the model experiment. Hence, our argumentation is always based on the similarity of inter-model correlation and the model experiment. If both show similar patterns, also the mechanism is expected to be similar. In the second paragraph of the revised discussion we try to explain this line of argumentation more clearly and also clearly state that the results for CMIP6 are not a strict proof but more a hypothesis: "Based on similarity between CMIP6 inter-model correlations with the effects of the Pacific moist bias in the EMAC model experiment, we hypothesize in the following that CMIP6 model differences could partly be related to the moisture content in the Pacific UTLS."

Furthermore, we also see that for the Pacific zonal wind the correlation is somewhat weaker and the patterns show larger differences. We added the new Fig. 3 e–f to show the model correlation in the relevant region (tropopause region above the Pacific) even clearer. Also, we included a cautious note when discussing the figure: "Furthermore, CMIP6 inter-model correlations between zonal wind and Pacific UTLS moisture show a significant correlation in the Pacific subtropics around the tropopause (Fig. 3 d, f), although weaker than for meridional wind at the eastern monsoon edge (Fig. 3c)."

Analysing inter-annual variability further could be interesting, but would optimally need longer simulations than the 20 years we have available, and also the existence of many other additional variability factors will likely obscure a clear picture. Instead, we present the correlation for wind velocities averaged over a larger region in the new Fig. 3 e/f. Although also this correlation is not too compact, it is clearly significant, and we use it to discuss the applicability and limits of the approach as well as effects of certain model characteristics, as suggested by the other reviewers.

*2) While the role of SWV can be indeed nicely explained, the role of other radiatively active species in the stratosphere is a lot less clear. Among them, ozone is another major heating source in the tropical stratosphere, but it's not discussed at all. I would expect the implementation of CLAMS to also affect the ozone in e.g. indirect ways. Have the authors looked into changes in ozone between the regular EMAC and EMAC-Clams? Would these be big enough to also play a role in the differences seen in terms of the large-scale circulation?*
Considering other trace gas species besides water vapour is definitely a very good suggestion. However, so far the coupling of Lagrangian transport in EMAC to radiation and dynamics only works for water vapour. Extending this approach to other chemical tracers (e.g. ozone) is the focus of ongoing model development work. On the other hand, it could also be seen as an advantage of the current simulation set-up for the present paper, that it allows the effects of stratospheric water vapour to be unambigously isolated, without interference with the effects of other trace species. To briefly mention these aspects we added a short sentence about the effects of other trace species besides water vapor at the end of the revised discussion: "Differences in model transport could also affect other trace gas species with large gradients, like ozone. However, as water vapour shows particularly steep gradients in the UTLS also the associated effects on circulation are expected to be comparably strong."

*3) While the role of the westerly wind duct is clear (at 100 hPa) in linking SWV biases and the anticyclone circulation, the role of other (prominent) dynamical features of the lower stratospheric circulation are less clear... such as, for example, the QBO jets, the cold point tropopause, the tape recorder, etc. I would recommend the authors to give a "broader" view of the effects of the implementation of CLAMS, aside from the localized effects at 100 hPa.*
As explained in the introduction (last paragraph) the focus of the present manuscript is on regional water vapour biases and regional circulation effects. Global effects on the zonal mean state of lower stratospheric water vapour changes have been discussed in the recent paper by Charlesworth et al. (2023), including effects on subtropical and eddy-driven jets, stratospheric circulation upwelling (related to the tape recorder) and the cold point tropopause. We precised the related text in the revised manuscript so that these issues become clearer (in the second paragraph of the introduction): "Charlesworth et al. (2023) showed that lower stratospheric water vapour, in particular the model moist bias in that region, exerts a first order effect on the zonal mean atmospheric circulation, with water vapour increases causing a strengthening of the stratospheric circulation, upward and poleward shifts of the subtropical jets, and a poleward shift of the tropospheric eddy-driven jet."

**Specific comments:**

L4: *I'd recommend adding a specific altitude range when talking about "regional circulation systems" (this also applies to L8).*
We added "...regional circulation systems in the UTLS" to make the altitude range clear.

L132: *I'm not entirely convinced about the causality... as many things change across different CMIP6 models. What about, for example, the role of vertical resolution across them?*
We fully agree with the reviewer here that also other factors likely play a role besides UTLS water vapour. Therefore, we chose a weak formulation here: "overestimated westerly ducts... appear to be related, at least partly, to the model moist bias ...". In the following, new paragraph in the revised version , related to the new subfigure Fig. 3 e–f we also briefly discuss resolution effects (see general comment (ii) above).

L143: *What about the effects of ENSO on the water vapor? WOuld that relationship also change in the CLAMS version*

*of EMAC?*
This is a good question. ENSO substantially affects the moisture budgte in the UTLS and it could well be that with a different representation of moisture transport in the model (as implemented here) also the simulated transport effects due to ENSO change. Focussing on inter-annual variability would be an interesting topic for future work, but would also require to extend the simulation length beyond the 20 years considered here.

Figure 4: *Would it be possible to also see the lines for EMAC and EMAC-CLAMS in this figure?*
We agree that it would be helpful to include some comparison between the EMAC model experiment and the CMIP6 simulations. However, these different model simulations are not exactly comparable as the EMAC experiment doesn't use exactly the same configuration as the historical CMIP6 simulations considered (as described in the appendix). Hence, a too detailed comparison between the EMAC experiment and CMIP6 could be misleading.
Nevertheless, we include the relation between $u$- and $v$-wind speed and the Pacific UTLS moisture from the two EMAC simulations (control and modified–Lagrangian) into the new figure Fig. 3 e–f together with the relation for CMIP6 models. On the one hand, this figure shows that also for the modified-Lagrangian simulation the agreement with the observaions is not perfect. On the other hand, the difference between the two EMAC model versions agrees well with the mean relation for CMIP6 (black dashed line), providing additional support that a similar mechanism is at work.

*Why is 250 hPa the lower boundary chosen for CLAMS? Are results sensitive to this choice?*
The level 250 hPa (more precisely, the model level closest to 250 hPa) as the lower boundary for the Lagrangian water vapour calculation has been chosen to ensure that it is only stratospheric water vapour that differs between the control and the modified–Lagrangian simulations. In pricinple, any level could be chosen, and as long as that level is below the tropopause the results for the lower stratosphere will be largely insensitive. In the model set-up used here, the Lagrangian transport calculation also extends all down to the surface, but below 250 hPa it is overwritten with the EMAC control water vapour field in each time step. This guarantees that both simulations have exactly the same water vapour field in the troposphere below about 250 hPa.

*General recommendation for 5 Appendix → I think this info should be moved into the main text, as lots of essential information is "packaged" into the Appendix, Since there are no length limitations for this journal that I'm aware of, I'd strong recommend restructuring and move all this info into the main paper.*
We understand that for ACP–Letters it is required to have all data and method information in the Appendix. Also, there is a length limit of 2000 words for the main part of the manuscript. Hence, we stay with the structure as is.

*General comment: while the impact of the diffusive transport scheme is clear and nicely demonstrated, it would be nice if the authors could comment on the role of other features on SWV, such as convective overshooting. Are there any changes in the Monsoon Anticyclone that are also driven "from the troposphere" or do all the differences only originate in the stratosphere?*
We totally agree that in general the UTLS moisture budget is affected by many different transport processes, including advection, mixing, convection, etc. In the control EMAC simulation in the UTLS, however, the water vapour distribution is largely controlled by advection and dehydration. This was shown from an additional EMAC sensitivity simulation with all water vapour tendencies output, as plotted in supplement Fig. 2. At least in EMAC at these levels the contribution of convective transport is minor. These sensitivity results are discussed in the last paragraph of the "EMAC model simulations" appendix section. Related to the reviewer comment we added the note: "This EMAC sensitivity analysis does not exclude a more significant effect of convection in other models or in the real atmosphere."

---

## Author Comment (AC3)

**Reply to Reviewer 2**

We thank the Reviewer for the careful reading and evaluation of the manuscript and the good comments which helped a lot to further improve the paper. In the following, we address all comments and questions raised (Reviewer's comments in italics). Text changes in the manuscript are highlighted in color (except minor wording changes).

Besides several specific comments, we see two common main concerns raised by both Reviewer's, regarding (i) the presentation quality and clarity of figures, and (ii) the discussion of specific climate model characteristics. A short overview of the related changes in the revised manuscript is:

(i) To enhance the **presentation quality and clarity** we modified all figures in the revised manuscript. In particular, we chose different color schemes, reduced the number of contours, changed to difference plots for certain quantities and improved the model-correlation visualization (a detailed list of figure changes is in the reply to the specific comment below). We are confident that these changes clearly improve the presentation of our results.

(ii) To relate our results to specific **model characteristics**, we included a new subfigure Fig. 3 (e, f). This figure shows the inter-model correlation for wind velocities averaged over the region of maximum signal, and includes information on horizontal and vertical resolution of the models by highlighting models with relatively high number of horizontal grid points with horizontal lines and models with high number of levels with vertical lines. Based on this simple model classification, no clear relation of resolution differences to the simulated water vapour bias is found. These findings are briefly discussed at the end of the discussion section and described in more detail in the Methods section 5.1.5.

A more detailed reply to all comments and description of the changes in the revised manuscript is given below.

**General comments:**
*Ploeger et al. explore the connection between lower stratospheric water vapour anomalies and dynamical biases previously identified by Charlesworth et al. in CMIP6 models using a combination of CMIP6 model output, observations, and specifically designed simulations using the EMAC model run with different transport schemes. Their finding that Pacific UTLS water vapour anomalies impact regional circulation builds on the findings of Charlesworth et al. by exploring inn more detail local water vapour anomaly-circulation bias connections, particularly with relation to the Asian summer monsoon and transport from Asia into the Pacific. I found the manuscript to be well written and the analysis clear. I feel the paper fits the scope of ACP and explores and important topic. However, I would recommend the authors address the comments below before publication.*
Thanks for this overall positive evaluation of the manuscript!

*Care should be taken throughout the paper to make a better distinction between water vapour anomalies and biases. To me, the anomaly is the relative abundance of local water vapour with respect to the zonal mean, whereas a bias is specifically a comment about the local abundance of water vapour in the model vs observations. These quantities may be related to an extent, but it is possible to imagine a very dry model that has a large water vapour anomaly over the Pacific having a small bias with respect to observations. I feel throughout the paper these two terms are used interchangeably, and this should be addressed.*
The use of the terms "anomaly" and "bias" in the paper should exactly be as the Reviewer here suggests. "Anomaly" is the relative enhancement of water vapour with respect to the zonal mean, while "bias" is the deviation of simulated water vapour from the observations. We added a clarifying sentence in the introduction and looked carefully through the entire manuscript again to ensure that this terminology is used consistently throughout the paper.

*In their analysis the authors use different lengths of time, over different years, for the different datasets used in this study. The EMAC model data used for analysis covers 2000-2009, the CMIP6 model data covers 2000-2014, and the ERA5 data covers 2005-2015. Given that the authors speak about the variability of water vapour on decadal timescales in the opening sentences of the introduction, I wonder to what extent the features identified in the paper are dependent on the choice of these relatively short time periods. For the CMIP6 data it is conceivable that averaged over multiple ensemble members 15 years represents something of a robust climatology, but can the same be said of 10 years of ERA5 or EMAC data? If the authors had used a different time period (e.g., 2000-2009), or a longer time period (2000-2020), would the ERA5 data show the same results? Can the authors say anything about this that strengthens the arguments made in the paper?*
We checked the robustness of the water vapour and circulation responses in the EMAC model experiment by plotting Figs. 1 and 2 also for the 10 years before (1990-1999), which still guarantees 10 years of model spin

up after switching to the Lagrangian transport scheme (see Methods 5.1.2). Regarding ERA5, we checked the robustness of the climatology by switching to the period 1990-1999 and 1979–2020. The figure below in the response letter (Fig. 1) shows that all results are insensitive to the choice of the period (for ERA5 only 1990– 1999 is shown, but the figure looks almost the same for 1979–2020). A short statement on this robustness is included at the end of the first paragraph in the Methods section 5.1.2.

*Many of the figures are difficult to read and crowded with information. Figures 1 and 2 have a lot going on, and it is difficult to see the black contours and arrows on the very dark red and blue shading. In figure 3 it is very difficult to tell the values form the blue and red shading. I would recommend replotting many of the figures to improve readability, and where possible separating the figures into more panels.*

Thanks for this very helpful comment, which was similarly brought up also by the other Reviewer! We agree that the clarity and readability of many figures should indeed be improved. For the revised manuscript we did significant efforts to improve the presentation quality. The changes to the figures are summarized below:

- We reduced the number of contours plotted by removing redundancy and focussing only on the most relevant variables. Therefore, PV contours are not shown in Fig. 1 a–d anymore, but only in Fig. 1 e–f. Here, we now show PV together with the zonal wind distribution at 100 hPa in the two EMAC model simulations to illustrate the circulation differences more clearly. Figure 2 is even more substantially reorganized, by showing the observation and Lagrangian EMAC–CLaMS model water vapor and PV fields together with the model difference between the two EMAC simulations (Fig. 2 c, f). The water vapor plots also show temperature contours (and only these), while the PV plots now show zonal wind contours (model differences in Fig. 2 c, f). With this reorganization of figures still the relevant variables are shown, but with much more clarity.

- The color scheme for anomalies and differences is changed to schemes with somewhat lighter maximum blue and red colors, so that dark contours are still readable. Also contour colors have been changed to have less different colors and still ensure optimal contrast (e.g. only black contours in Fig. 1, only white contours in Fig. 2, tropopause as grey lines in Figs. 2 and 3).

- Circulation differences between the two EMAC model simulations (control vs. modified Lagrangian) are now shown in difference plots in Fig. 2 c, f. These new subfigures show the temperature, zonal wind and PV differences as induced by the stratospheric water vapor differences with much more more clarity compared to the manuscript version before. In particular, the cooling effect as well as the strengthening of PV gradients of the enhanced lowermost stratospheric water vapor can be clearly seen now.

- Regions of relevant temperature gradient changes explaining the wind changes in Fig. 3 a, c are now highlighted with white hatching. Although not including quantitative information on the exact values anymore, the modified figure still shows qualitatively that the relation between vertical wind shear and horizontal temperature gradient changes is consistent with thermal wind balance.

We think that with these changes the presentation of our results has become much clearer.

*It is clear from the CMIP6 multi-model mean and figures S3/S4 that the Pacific water vapour anomalies are common across CMIP6 models. However, not all models show large positive anomalies with respect to the zonal mean above the region of the Asian summer monsoon (most notably the CNRM model, some configurations of the CESM model, and the SAM0-UNICON model). Can anything be inferred about links between Pacific water vapour anomalies and regional circulation biases in these models that look quite different to the others? Going further, can anything be said about model structural differences (e.g., resolution, transport scheme) in driving the biases explored in this study, or is it the case that whatever the model resolution, global models are too susceptible to processes like numerical diffusion? I think I'd like to see some discussion exploring model differences in the manuscript.*

We like the idea to investigate in more detail whether particular CMIP6 model differences can be related to the development of the moisture bias. However, as explained also in the response to the second comment of the other Reviewer (see also main change (ii), summarized in our general comment above) this question is not easy to answer with the available suite of simulations. As also formulated by the other Reviewer, one interesting question here is whether it is the horizontal or vertical resolution which is most critical for the moisture biases. The water vapor maps for single models in the supplement (Fig. S3) show that indeed the model with highest resolution (MPI-ESM1-2-HR) simulates a well confined monsoon moisture anomaly and only a weak Pacific moisture bias. However, the differences in water vapor bias to other models are not related in a simple manner to differences in vertical or horizontal resolution. For instance, models with either a comparable number of vertical levels (e.g., MRI-ESM2-0) or with a comparable number of latitude/longitude grid points (e.g., BCC-CSM2-MR) show stronger moisture biases than the model with highest resolution (MPI-ESM1-2-HR). Furthermore,

as already said by the Reviewer, regarding vertical resolution it is likely the number of levels in the UTLS rather than the total number of levels, and information on that is not well documented. In general, as the transport schemes employed by the different models are quite different, the differences in the simulated water vapor distribution are not simply explanable by differences in the number of vertical levels or horizontal grid points. To discuss these aspects in the revised manuscript, we included a new subfigure Fig. 3 (e–f) which shows the inter-model correlation between Pacific $u$- and $v$-wind with the Pacific water vapor index for the winds averaged over the UTLS region of most significant correlation (see figure caption and Methods). In addition we selected from the CMIP6 models a few models with relatively high horizontal and/or vertical resolution (indicated by the horizontal/vertical lines in the scatter plot). The criteria for selecting these models are given in the Methods section 5.1.4. Clearly, neither differences in horizontal nor in vertical resolution explain the spread of simulated Pacific moisture bias. Hence, a more detailed analysis of model differences, best including sensitivity simulations with the same model and differing horizontal/vertical resolution would be needed to draw robust conclusions on this issue. We discuss these points in the revised manuscript in a short paragraph at the end of the discussion section and in the Methods section 5.1.5.

**Specific comments:**

L13-14: *"Variations in stratospheric water vapour have been shown to modify past global warming by up to 30% (Solomon et al., 2010)" This sentence is misleading – the Solomon et al. paper says that variations in lower stratospheric water vapour on a decadal timescale can have a significant radiative impact, but this sentence could be read as 30% of past climate change can be attributed to water vapour changes, which is not the case.*
Thanks for pointing this misleading formulation out (as similarly by the other Reviewer)! We changed the sentence to: "Decadal variations in stratospheric water vapour have been shown to modify the radiative budget on a decadal scale by up to 30%."

L19: *My understanding is that the 0.1-0.26 $Wm^{-2}K^{-1}$ range from Banerjee et al. is calculated using CMIP5 models. Is there an estimate from CMIP6 models? I know that a recent paper by Nowack et al. (2023) has tried to constrain this estimate and comes up with a slightly reduced range of 0.086-0.201 $Wm^{-2}K^{-1}$, which may be worth noting here.*
We agree that it would be more appropriate to refer also the most recent feedback estimates. We modified the sentence accordingly and include the Nowack et al. (2023) reference in the revised version.

L36-38: *"Analysis of the distribution of water vapour in the lower stratosphere simulated by a climate model has shown a substantial model bias in the Asian monsoon region" I was not sure here if the authors are talking about analysis of a single climate model, or making a statement that is true for all models.*
This sentence refers to the results of Wang et al. (2018) which are based on a single climate model (WACCM). The sentence is slightly modified to make clear that it refers to a "specific climate model".

L52: *Make it clear the anomalies are with respect to the zonal mean distribution of water vapour.*
We added "as compared to the zonal mean distribution" to make the terminology here more clear. (See also our reply to the first comment regarding the use of a consistent terminology).

L63-64: *Here, when saying the EMAC anomaly is more pronounced than the CMIP6 multi-model mean, I feel it is worth explicitly stating whether the EMAC model is an extreme case, or within the CMIP6 model spread, citing figure S3/S4 as evidence.*
We agree that it is worth noting that the EMAC moist bias lies within the CMIP6 spread, and added this information to the sentence: "..., but overall lies well within the spread of CMIP6 models (Supplement Fig. S3)."

L71-72: *When speaking about temperatures here, it would be great if the authors could show the temperature difference between EMAC and EMAC-CLAMS in a separate figure/panel (especially given general comment above about figure readability).*
We totally agree with the Reviewer that presenting difference plots would enhance clarity. Temperature differences between control and modified–Lagrangian EMAC simulations along with water vapour, PV and zonal wind differences are now shown in the new Fig. 2c. (See also our reply to the above general comment about presentation and figure quality).

L114-116: *"The CMIP6 inter-model correlation between local meridional v–wind velocity in the monsoon anticyclone and a water vapour index measuring the strength of the Pacific moisture anomaly (Methods) shows a significant anticorrelation eastward of 100∘E (Fig. 3d)." I am unsure here what feature I should be looking at. Firstly, I think this should be*

*figure 3c, not 3d. If so, is it the statistically significant blue shading between 100-50 hPa at around 150 degrees? Please provide more specific description in the text. Additionally, it is very hard to read from the figure the correlation in this region. Perhaps it could be given in the text?*

There was indeed a typo in the referencing and the correct figure-reference here should be Fig. 3c. Thanks for noticing that! As the Reviewer assumes, this sentence refers to the negative correlation around 150 degrees E between about 100-50 hPa. We modified the description in the text to make this more clear, and also think that the improvements made to Fig. 3 (see reply to the above general comment about figure quality), in particular less color contours, help to enhamce readability and clarity. In addition, we included scatter plots for the correlation in the relevant region as the new Fig. 3 e–f, which shows the inter-model correlations in the relevant region even clearer.

L119-120: *"Above the tropopause, this correlation pattern closely resembles the pattern of differences in meridional flow (v–wind) 120 between the control and modified–Lagrangian simulations in the EMAC model experiment (Fig. 3a)" Does it? There is some overlap in the blue shaded region in figure 3a and 3c, but the features themselves have very different shapes, horizontal extents, and extensions into the troposphere.*

We agree that in particular the downward extension of signals into the troposphere shows clear differences between the EMAC model experiment and the CMIP6 ensemble. However, above the tropopause in the lower stratosphere the pattern of negative/positive differences agrees remarkably well with the pattern of negative/positive inter-model correlation, in particular given the fact that also other factors than stratospheric water vapour affect the UTLS circulation in CMIP6 models. We modified the text slightly to make even more clear that we refer here to the signal in the lower stratosphere.

**Technical corrections:**

L62: *remove 'also'*
Done.

L166: *oxydation should be oxidation*
Done.

Figure caption for S4: *I believe it should say (Fig. S3 continued)*
Done.

[Figure]

Figure 1: Sensitivity of manuscript figures 1 and 2 to the choice of climatological period, by showing all distributions for the 1990–1999 period (except g). (top) Water vapour zonal anomalies at 100 hPa in boreal summer (June–August) from modified–Lagrangian (left) EMAC–CLaMS and (right) EMAC. (middle) PV cross-sections in the Pacific region (180°-200°E) in boreal summer (June–August), (left) ERA5, (middle) modified–Lagrangian EMAC–CLaMS, and (right) the difference between control EMAC and modified EMAC–CLaMS simulations. (bottom) Correlation between water vapour and potential vorticity at 360 K potential temperature level in the Pacific region (15°-70°N, 140°-240°E) for (g) MLS satellite observations and ERA5 reanalysis PV (here for 2005–2015), (h) control EMAC, and (i) modified–Lagrangian EMAC–CLaMS. The Pearson correlation coefficient $r$ and linear regression slope $a$ are given in each figure and the linear regression fit is illustrated as black dashed line.